# Study of Monitoring Method and Melt Flow Behavior in Compression Molding Process Using Thermoplastic Sheets Reinforced with Discontinuous Long-Fibers

**Masatoshi Kobayashi**

Innovative Research Excellence, Honda R&D Co., Ltd., Tochigi 321-3321, Japan; Masatoshi_01_Kobayashi@jp.honda; Tel.: +81-80-9448-8485

**Abstract:** In compression molding using glass-fiber-mat-reinforced thermoplastic (GMT) sheets, a slightly longer compression waiting time from sheet placement on a lower mold to the start of sheet compression by an upper mold can cause incomplete filling due to a decrease in the sheet temperature. However, precise measurement techniques for compression waiting time have not been sufficiently established. A monitoring system was produced that includes pressure—temperature sensors mounted in a compression mold that can simultaneously measure the pressure and temperature of one local surface. Two types of distance sensors were also used to measure upper mold motion widely and precisely. Determination of compression waiting time was attempted by measuring the moment when the lower mold temperature slightly increases in response to contact with the melted GMT sheet and the moment when the melt pressure increases in response to compression by an upper mold. The results showed that compression waiting time could be precisely calculated using the profile data obtained. Moreover, it was also possible to observe the melt pressure overshoot that occurs depending on sheet stacking patterns and mold cavity shape, although in some cases, the overshoot was not observed. In conclusion, this study has demonstrated that the system is effective in monitoring the compression molding process widely and precisely.

**Keywords:** glass-fiber-mat-reinforced thermoplastic; polyamide 6; stacked sheet; compression molding; melt flow; in-mold measurement; pressure–temperature sensor; compression waiting time; pressure overshoot

## 1. Introduction

In the automotive industry, lighter materials such as glass-fiber-reinforced plastics and carbon-fiber-reinforced plastics have been applied to vehicle parts for improvement of fuel efficiency. Currently, vehicle parts made from fiber-reinforced plastics (FRP) with excellent specific strength, specific rigidity, and good moldability are also being constantly developed, e.g., [1–10]. In the case of the mass production of parts reinforced with discontinuous long fibers more than 10 mm long, a compression molding method using sheet-shaped fiber-reinforced thermoplastics is generally applied. As a mass production method, the compression molding process has an important role in the automotive industry.

The two main groups of sheet-shaped thermoplastics reinforced with discontinuous long fibers used in compression molding are glass-fiber-mat-reinforced thermoplastics (GMT) and long-fiber-reinforced thermoplastics (LFT) [1–3]. GMT sheets are manufactured through a process of impregnating fibers with resin without screw mixing, so that longer fibers than the LFT remain, and most fiber bundles are not separated. This molding method for GMT parts does not require special equipment, and a general purpose heating machine and a general purpose compression molding machine are the minimum pieces of equipment needed. On the other hand, LFT sheets are manufactured through a process of impregnating fibers with resin using screw mixing, so that shorter fibers than GMT remain and fiber bundles are separated. Normally, sheet-shaped LFT sheets are used in a molding

method called the LFT-D, in which parts are produced at the same time as materials are manufactured. The molding method for the LFT parts requires special equipment. Parts suppliers, who prefer to prioritize fiber length and use general purpose equipment owned, without purchasing expensive specialized equipment, can also choose GMT. Figure 1 shows a typical compression molding process procedure for products using GMT sheets.

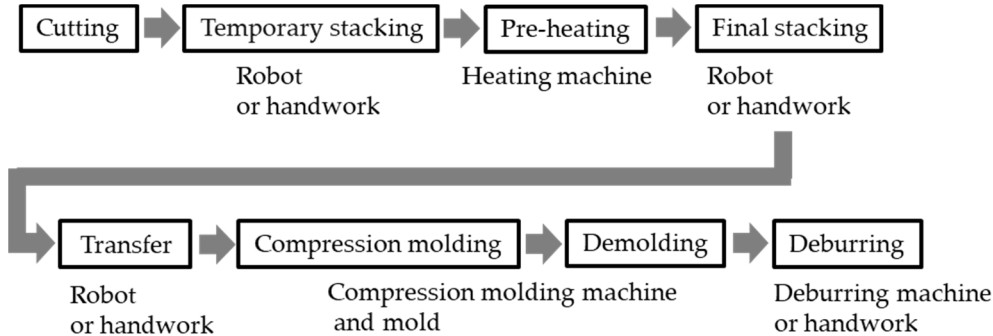

**Figure 1.** Typical compression molding process for glass-fiber-mat-reinforced thermoplastic (GMT) parts.

GMT sheets, which are reinforced with discontinuous long fibers randomly oriented in the plane, are molded using a non-isothermal compression process with heating and cooling. Here, the compression molding process for GMT sheets and its issues are outlined. First, GMT sheets with adjusted dimensions are prepared. In many cases, multiple sheets are stacked. The sheets are heated to a temperature above the melting point in a heating machine and are then compressed using an upper mold immediately after being placed on a lower mold, which is controlled so that it is below the crystallization temperature. At this time, the viscosity of the melted GMT, which flows while cooling due to contact with the mold surface, increases significantly, and moldability changes rapidly over time. In general, GMT sheets have higher thermal conductivity than non-reinforced thermoplastics since highly concentrated fibers (e.g., 40 wt.% or more) are contained. This is one factor in the significant viscosity increase. Due to this sudden change in moldability, especially for parts with complex shapes, many engineers have to spend a great deal of time narrowing down appropriate molding conditions that can completely fill a mold cavity, through trial and error. In order to completely fill a mold cavity with melted GMT, it is preferable to complete the flow while the viscosity is still sufficiently low. Therefore, melted GMT sheets removed from the heating machine are transferred to a lower mold as soon as possible. It is not uncommon for transfer time to take more than 10 s, even with automated equipment in the actual production process. Immediately after melted GMT sheets are placed on a lower mold, many engineers operate molding machines under setting conditions in which the upper mold can be lowered as quickly as possible. At this time, sheet temperature, mold temperature, compression load, compression speed, and compression waiting time are important setting conditions for complete filling. Although it is necessary to judge whether or not setting conditions are appropriate from data measured using various sensors mounted in a heating machine, a molding machine, and a mold, it can be difficult to measure compression waiting time precisely. It is not uncommon for compression waiting time to take more than 10 s when using a general purpose medium-sized or large-sized compression molding machine. Compression waiting time here refers to the time difference between when a heated sheet is placed on a lower mold and when a heated sheet begins to be compressed by the downward movement of an upper mold. A shorter compression waiting time provides a better chance to fully fill a mold cavity since the GMT viscosity increase due to the temperature decrease is inhibited. That is to say, a slightly longer compression waiting time can turn a complete filling (i.e., acceptable condition) into an incomplete filling (i.e., unacceptable condition). Therefore, the actual compression waiting time is one of the process parameters that must be controlled and monitored when production parts are manufactured. Molding conditions are set using

the operation panel on molding equipment. Profile data for some parameters, excluding actual compression waiting time, are easily logged using the normal monitoring system that comes with molding equipment. However, actual compression waiting time does not depend only on the capacity of a compression molding machine, such as the platen moving speed and responsivity, but also on the moving speed of a transfer robot or a worker and the upper mold position during standby. Machine operational variations peculiar to hydraulic systems must also be taken into consideration (e.g., the variations caused by oil viscosity change due to oil temperature increase during repeated operations and due to ambient temperature change). When multiple stacked GMT sheets are placed on different positions of one lower mold using one transfer robot or one worker, the actual compression waiting time for each sheet can also be different. In addition, not just melt pressure and temperature but also actual compression waiting time is desired as data that can be managed and stored for statistical process control (SPC) and product traceability, which ensures the quality of production parts at many factories. In other words, measurements of actual compression waiting time are required every time a product part is molded. Thus, even though compression waiting time is very important in industry, appropriate measurement methods have not yet been sufficiently established.

Next, melt pressure during compression is also an important process parameter that should be controlled. Although it is preferable to apply as high a melt pressure as possible to melted GMT for complete filling, excessively high melt pressure reduces mold durability and causes breakage. Therefore, most engineers always pay attention to whether melt pressure is applied within an appropriate range. When production parts are molded, GMT sheets are very often stacked before compression. That is to say, the stacked GMT sheets frequently flow in a mold cavity. Nevertheless, the influence of the sheet stacking pattern on melt pressure has not been sufficiently examined. In comparison with a mass of inseparable LFT (e.g., bulk LFT), a melted sheet that is stacked has boundary surfaces between each ply with very low thermal adhesion. There is a possibility that the boundary surfaces cause a unique melt flow behavior in a mold cavity, and melt pressure profile data may be influenced by its behavior.

Many earlier articles have already reported on examining melted polymeric material behavior in mold cavities using in-mold sensors in injection molding, e.g., [11–13]. However, there are few reports on examinations of melted GMT sheets that are stacked in a mold cavity using in-mold sensors in compression molding [1]. As a basic model experiment, instead of a compression mold and a compression molding machine, simple disk plates and a tensile compression testing machine with a constant-temperature oven that is used to measure specimen characteristics such as load–displacement curves were used. The GMT sheets were compressed under isothermal conditions, and then time-dependent changes of compressive load obtained from a load sensor were examined [14]. As molding experiments using a compression mold and a compression molding machine, there are several reports discussing time-dependent changes in melt pressure and temperature for GMT under non-isothermal conditions through a compression mold with pressure and temperature sensors mounted at different positions. In these reports, the experimental data were compared with numerical simulation results [15], and melt pressure reductions, which are due to material shrinkage by temperature decrease after indicating maximum pressure, were discussed [16]. However, profile data on melted sheet transfer from a heating machine to a mold have not been measured, and measurement techniques for precise "compression waiting time" have not been sufficiently verified. The influence of the sheet stacking pattern on melt pressure has also not been examined.

In this article, molding experiments were conducted under non-isothermal conditions using a heating machine for pre-heating, a compression mold, and a medium-sized industrial compression molding machine. An in-mold monitoring system, including pressure—temperature in-mold sensors mounted in a compression mold, was also constructed. A pressure—temperature in-mold sensor is able to simultaneously measure pressure and temperature on one local surface. The system was also connected to a tem-

perature sensor mounted inside the heating machine for pre-heating, two distance sensors for the upper mold position measurement, and a compression load sensor for an upper mold position measurement. Therefore, it was possible to simultaneously monitor the temperature and melt pressure of a stacked GMT sheet placed on a lower mold the furnace temperature in the heating machine, in addition to the machine movements. Compression molding tests were performed using glass-fiber-mat-reinforced polyamide 6 sheets that were stacked as GMT, and a large amount of profile data was logged by the system. Through the profile data obtained, the author attempted to estimate a precise compression waiting time and examined melt flow behavior by comparing the melt pressure data of GMT sheets with different stacking patterns. The results showed that the system was able to monitor a wider process from pre-heating start to demolding of a molded part after compression. Compression waiting time could be precisely determined. Moreover, melt pressure overshoot that occurs depending on sheet stacking patterns and mold cavity shape could be observed. The mechanism that causes the overshoot has been discussed.

## 2. Material

GMT sheets with a nominal 1.5-mm thickness made from discontinuous long E-glass fiber-reinforced polyamide 6 were used. The sheets were prepared in two shapes: one a square 290 mm wide and 290 mm long and the other a rectangle 145 mm wide and 290 mm long. The glass fibers were already fully impregnated with the polyamide 6. The weight fraction of the glass fibers was 67 wt.%. The fiber bundles, from 30 to 50 mm in initial length, were dispersed in mat state and were randomly oriented in a plane. Steady viscosities measured at 240 °C using a rotary rheometer (ARES, TA Instruments, New Castle, Delaware, USA) and a parallel plate fixture were 7570, 2650, and 660 Pa·s at a shear rate of 0.1, 1, and 10 s$^{-1}$, respectively. Before molding, the sheets were hot-air dried for more than 12 h at 80 °C, in order to remove any moisture absorbed inside the polyamide 6.

## 3. Molding Equipment

As shown in Figure 2, the molding equipment consists of a heating machine for pre-heating and a compression molding machine with a mold attached. The heating machine was produced specifically for this study, and ceramic infrared heaters that can control both top and bottom individually in 9 zones are fitted in a furnace whose entire inner surface is surrounded by heat-insulating walls. The inside dimension of the furnace is 1370 mm wide, 3000 mm long, and 160 mm high. Furnace temperature is controlled by adjusting the electric power. The GMT sheets can be inserted into or removed from the furnace by opening a door. The sheets placed on a moveable wire mesh panel in the furnace are heated from above and below. In addition to a sensor for temperature control, another temperature sensor (k-type thermocouple), which has an excellent responsivity owing to a measuring junction part with a very thin flat shape (0.11-mm thickness), is mounted near the door in the furnace. Moments when the door is opened and closed can be judged through temperature changes measured using the very thin flat-shaped temperature sensor.

A hydraulically powered compression molding machine (FP-500, Amino Corporation; Fujinomiya; Shizuoka; Japan) with 4-axis slide guides was used. The stationary and movable platens both measure 2500 mm wide and 1500 mm long. Daylight, which refers to the vertical distance from stationary to movable platens, is 1300 mm in its maximum up position. The slide distance that the movable platen can move in the vertical direction is 1000 mm. The maximum compression force $F_{max}$ is 5000 kN. The maximum initial mold-closing speed when the movable platen descends is 200 mm/s. At mold positions lower than the speed-reduction position $H_{s\text{-}r}$, which is set to prevent mold damage due to the upper mold moving too fast, the mold-closing speed automatically decreases to a range from 0.1 to 60 mm/s, depending on the set compression force. Time-dependent changes in compression force can be taken from a load sensor originally built into the machine as one of the standard attached instruments. Time-dependent changes in the position of the upper mold can be taken from both a general long-distance sensor, originally mounted on

the movable platen as a standard attached instrument, and a high-precision short-distance sensor, which was mounted on the upper mold specifically for this study, with a red-light semiconductor laser at a wavelength of 655 nm (IL-S100, KEYENCE Corporation; Osaka; Japan). The former does not have high measuring accuracy but can measure long distances from the movable platen to the stationary platen, and it is used for measuring a wide range of positions from the beginning of the upper mold movement. The latter can measure only short distances (between 70 and 130 mm from the sensor head) but has a high measuring accuracy (repeat accuracy 4 μm), and it is used for measuring a narrow range of behavior from just before the mold is closed, along with a precision small jack with an adjustable height (from 64 to 86 mm) while maintaining very high parallelism within 100 μm (LJ-80133, SIGMA KOKI CO., LTD.; Tokyo; Japan). Changes in mold-closing speed over short distances during compression can also be estimated in detail using the data obtained from the latter. Incidentally, as a safety measure, a movable large-diameter support rod made of steel is set between the stationary and the movable platen in order to prevent the movable platen from dropping when an operator enters the stationary platen. The support rod is manually removed by an operator just before the movable platen descends when the upper mold compresses material for molding. The heating machine and the compression molding machine were positioned approximately 2 m away.

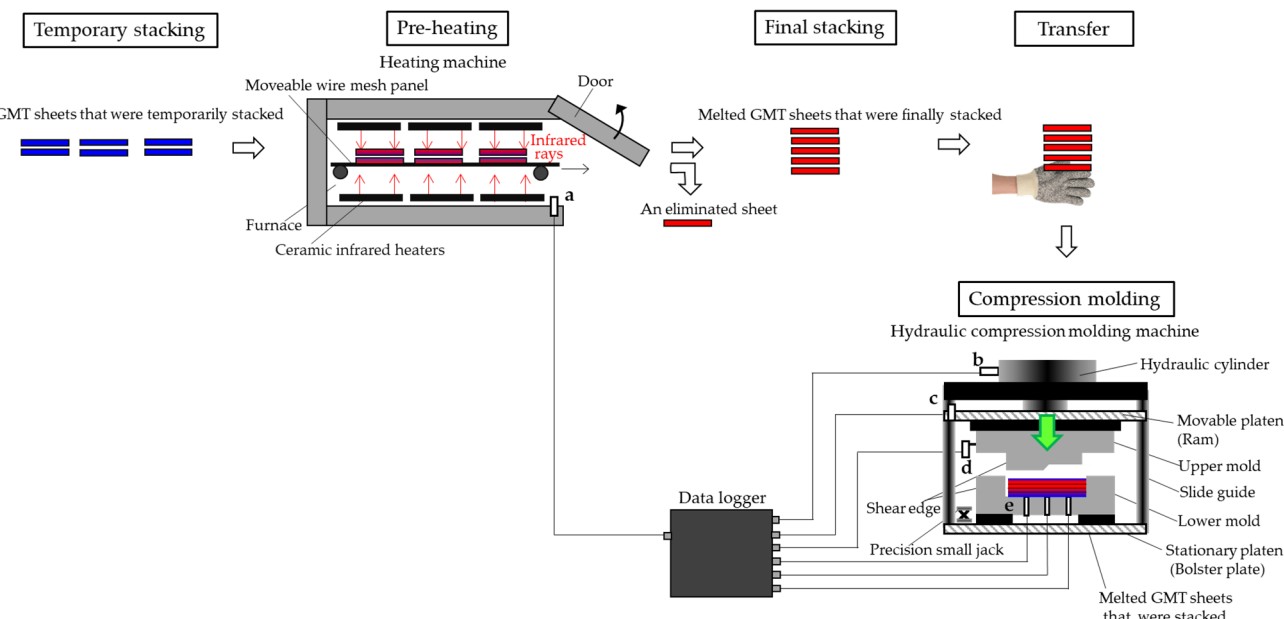

**Figure 2.** Schematic illustration of test equipment and monitoring system for compression molding in the case of stacking pattern A shown in Figure 5. (**a.** temperature sensor with excellent responsivity, **b.** load sensor, **c.** long-distance sensor, **d.** high-precision distance sensor, **e.** pressure–temperature in-mold sensors).

Figure 3 shows schematic illustrations of two types of compression molds made from S50C and the dimensions of two molded plates. Mold A, which has a flat surface on both the lower and upper molds, was produced for molding a uniform thickness plate. Mold B, which has a flat surface in the lower mold and a stepped surface in the upper mold, was produced for molding a non-uniform thickness plate with a thin section, a stepped section, and a thick section. These molds have a shear edge structure with a vertical wall height of 30 mm on four sides of the lower mold. That is to say, when the molds are closed, the cavity heights change from 0 to 30 mm depending on the position of the upper molds. The lower mold surface dimensions in the cavities of both molds are 300 mm wide and 300 mm long. Therefore, the width and length of the two molded plates are also approximately 300 mm each. Although molded plate thickness depends on the number of stacked sheets placed on the lower mold, when the stacking patterns described later were used, the uniform

thickness plate had a nominal thickness of 7 mm, and the non-uniform thickness plate had nominal thicknesses of 5.4 and 8.7 mm for thin and thick sections, respectively.

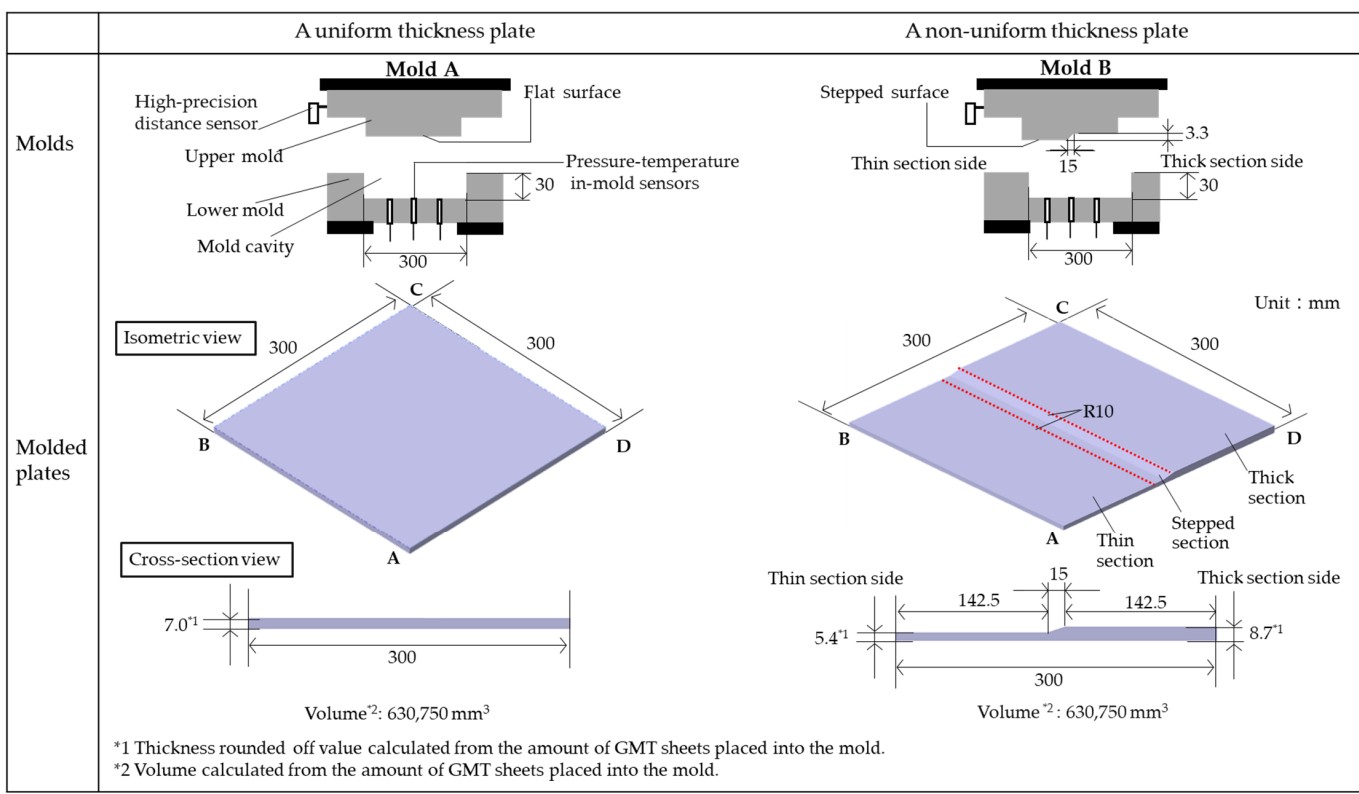

**Figure 3.** Schematic illustrations of compression molds and dimension of molded plates.

Figure 4 shows pressure–temperature sensor positions mounted in the lower mold. The pressure–temperature in-mold sensors with a flat front (Type-6189A, Kistler; Winterthur; Switzerland) were used so that pressure and temperature on the same local positions could be monitored simultaneously. Type-6189A is a pressure sensor in which quartz is used as a piezoelectric element, with a k-type thermocouple integrated. The sensors, which have a front diameter of 2.5 mm, were precisely mounted in three positions so that each flat front was flush with the lower mold surface. The small dimension results in a short response for not only pressure but also temperature measurements. Here, PT2, PT3, and PT4 are mounted at 102-mm intervals on the left, center, and right sides of the centerline, respectively. In the case of a non-uniform thickness plate, these sensors can measure pressure and temperature on the local surfaces in the thin section, the stepped section, and the thick section. As a broader molding monitoring system, in addition to the pressure–temperature in-mold sensors, two distance sensors for detecting the upper mold position, a load sensor for detecting compression force, and the very thin flat-shaped temperature sensor near the door in the furnace of the heating machine were also connected to one data logger.

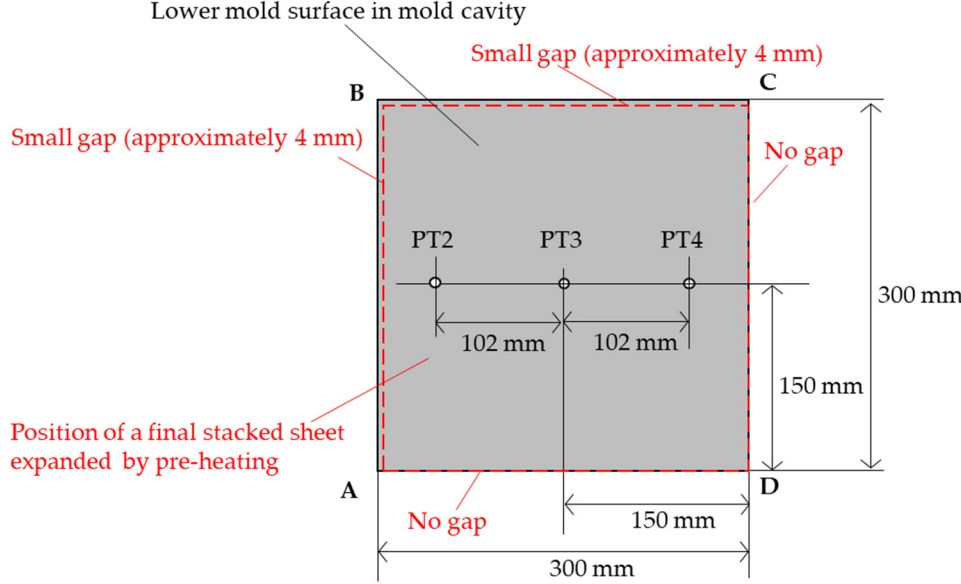

**Figure 4.** Positions of the pressure–temperature in-mold sensors and a final stacked sheet on the lower mold. (The corner symbols A, B, C, and D correspond to those of the molded plate shown in Figure 3).

## 4. Experimental Methods

All data obtained from the molding monitoring system were set to be logged at sampling intervals of 0.1 s. Table 1 shows the molding condition. The molding equipment described above was fixed in a temperature-controlled room. First, in the temporary stacking process, 3 sets of stacked GMT sheets, which were cut to the above-mentioned dimensions and temporarily stacked in 2 plies, were prepared at room temperature (i.e., a total of 6 sheets).

**Table 1.** Set molding conditions

| Condition Items | Symbol | Unit | Value |
|---|---|---|---|
| Room temperature | $T_{room}$ | °C | 20–25 |
| Furnace temperature | – | °C | 280 |
| Pre-heating sheet temperature | $T_{pre\text{-}heat}$ | °C | 270 |
| Pre-heating time | $t_{pre\text{-}heat}$ | s | 476 |
| Mold temperature | $T_{mold}$ | °C | 160 |
| Standby position of movable platen | $H_{standby}$ | mm | 1300 |
| Speed-reduction position of movable platen | $H_{s\text{-}r}$ | mm | 477 |
| Initial mold-closing speed (Initial platen speed) | $V_{initial}$ | mm/s | 200 |
| Final mold-closing speed (after passing $H_{s\text{-}r}$ ) | $V_{final}$ | mm/s | More than 0.1 (Automatically decelerations depending on $F_{max}$) |
| Maximum compression force | $F_{max}$ | kN | 3000 |
| Holding pressure time | $t_{holding}$ | s | 60 |

Second, in the pre-heating process, the door of the heating machine was opened, and the 3 sets of the temporary stacked sheets were placed into the furnace (Figure 2). As the number of stacked plies increases, stacked sheet thickness also increased, so that between it takes time to reach a uniform temperature [1]. Lengthy pre-heating time turns polyamide 6 a brown color, and there is concern about resin thermal degradation. On the other hand, if the sheets are put into the furnace one by one, it takes time for final stacking as the subsequent process, which leads to sheet temperature decreases. Therefore, temperature sensors (k-type thermocouple), which have an excellent responsivity owing to

a measuring junction part with a very thin flat shape (0.11 mm thickness), were attached to the surface and the center in the thickness direction of various stacked sheets using heat-resistant adhesive tape, and prior measurements of temperature differences between the surface and the center were performed as described in detail later. The author reduced the number of temporary stacking plies to only 2, in order to heat each stacked sheet to as uniform a temperature as possible in a short time. When the temporary 2-ply sheet was heated to 270 °C, which is sufficiently higher than the melting point (approximately 225 °C) of the polyamide 6, no rapid thermal degradation was observed in the furnace, which was controlled at 280 °C. At this time, the dimensions of the 2-ply square sheets thermally expanded from 290 mm wide and 290 mm long to approximately 296 mm wide and 296 mm long. The thermal expansion ratio was approximately 2.1%.

Third, in the final stacking process, the door was opened and the 3 sets of temporary 2-ply sheets, which were melted, were removed from the furnace when a required pre-heating time had elapsed. At that time, an operator manually prepared a final stacked sheet as speedily as possible. Here, Figure 5 shows the procedures for preparing the final stacking patterns. Usually, the GMT sheets are prepared by cutting them into square or rectangular shapes, even in mass production. However, many parts have various complex shapes. That is, an excessively large number of sheets may be locally placed on a thin area of a mold cavity. Conversely, an excessively small number of sheets may be placed on a thick area of a mold cavity. Therefore, stacked sheets in various patterns were prepared in this study. In pattern A, immediately after only one sheet was eliminated from the 3 sets of the temporary 2-ply sheets, a final 5-ply sheet, which consists of only square sheets, was prepared by the rest of the sheets. In patterns B, C, and D, each final 6-ply sheet was prepared from 2 sets of the temporary 2-ply sheets, which consist of square sheets, and 1 set of the temporary 2-ply sheet, which consists of rectangle sheets. With regard to the positions of the 5th and 6th layers counting from the bottom, those layers in pattern B were stacked on the side corresponding to the thick section side of the mold. Those layers in pattern C were stacked on the side corresponding to the center (including the stepped section) of the mold. Those layers in pattern D were stacked on the side corresponding to the thin section side of the mold. The amount of each material in all the final stacked sheets of these 4 patterns was the same.

Fourth, in the compression molding process, in order to inhibit the sheet temperature decrease as much as possible, the final stacked sheet was quickly transferred manually and placed in contact with the corner D of the lower mold controlled at 160 °C ($T_{mold}$). Although each of the final stacked sheets expanded with pre-heating, their widths and lengths were slightly smaller than those of the mold cavity. That is to say, as shown in Figure 4, the sheet placed was almost in contact with both side DA and side CD. However, there were small gaps (approximately 4 mm) between the sheet and side AB and between the sheet and side BC. Therefore, when the final stacked sheet was placed on the lower mold, the sheet occupancy rate for the lower mold area was approximately 97.4%. Here, the occupancy rate is calculated by dividing the sheet area by the lower mold area and multiplying it by 100; this is 100% when the material completely fills the mold cavity after compression completion. Under these conditions, the occupancy was already very high, even before compression. After placement, the operator stepped away from the mold as quickly as possible, removed the movable large-diameter support rod, which is a safety measure, and pushed the operation button of the molding machine. Immediately, the upper mold, which was standing by at a position $H_{standby}$ of 1300 mm, began to move down at an initial mold-closing speed $V_{initial}$ of 200 mm/s. After the movable platen on which the upper mold was mounted passed the $H_{s-r}$ of 477 mm, the $V_{initial}$ automatically decelerated and shifted to final mold-closing speed $V_{final}$, based on the set maximum compression force $F_{max}$ of 3000 kN. The time change of the upper mold position must be measured precisely in order to clarify the $V_{final}$ value because $V_{final}$ cannot be set directly by an operator and is also influenced by sheet viscosity.

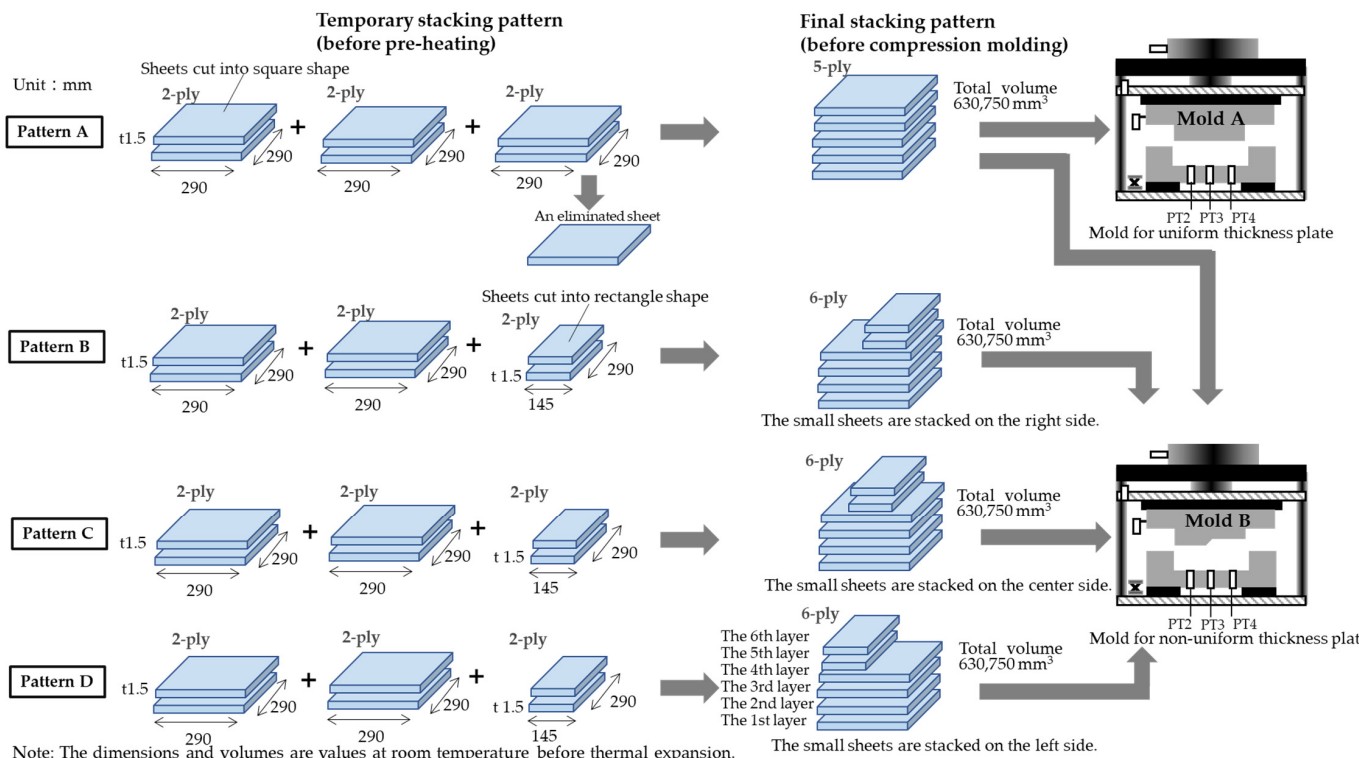

**Figure 5.** Final stacking patterns of GMT sheets.

Here, the time change of surface distance $D_m$ from the lower mold to the upper mold in the mold cavity was measured as follows. Figure 6 illustrates the setup of the high-precision short-distance sensor and the $D_m$ in the thin section side between the upper and the lower mold during the upper mold downward movement when mold B is used for a non-uniform thickness plate. As described above, this sensor can measure distances only in a narrow range (between 70 and 130 mm) because it is designed with a priority on measurement accuracy. That is, the high-precision distances begin to be detected when the sensor moves down to the position where the distance to the precision small jack is 130 mm. This position where detection can be started is defined as the upper limit measurement position $Z_{ulp}$. High-precision distances are no longer able to be detected when the sensor moves down to the position where the distance to the small precision jack is 70 mm. This position, where the detection cannot be performed, is defined as the lower limit measurement position $Z_{llp}$. Therefore, the height of the precision small jack must be adjusted so that the sensor head position can be slightly higher than $Z_{llp}$ when the upper mold moves down to the lowest position and compression is completed. $D_m$ can be calculated using the following equation from time-dependent data of the distance $Z(t)$ between the high-precision short-distance sensor and the precision small jack.

$$D_m = Z(t) - Z_{min} - th, \tag{1}$$

where $Z_{min}$ is a minimum value of $Z(t)$, and *th* is the thickness of a non-uniform thickness plate in the thin section side after molding. Understandably, $D_m$ can also be calculated using Equation (1) even when mold A is used for a uniform thickness plate. The final stacked sheet compresses at $F_{max}$ for a holding pressure time $t_{holding}$ of 60 s. Here, when the maximum compressive force is applied, the maximum compressive pressure calculated simply from the projected area of the plates (300 mm wide and 300 mm long) is approximately 33.3 MPa.

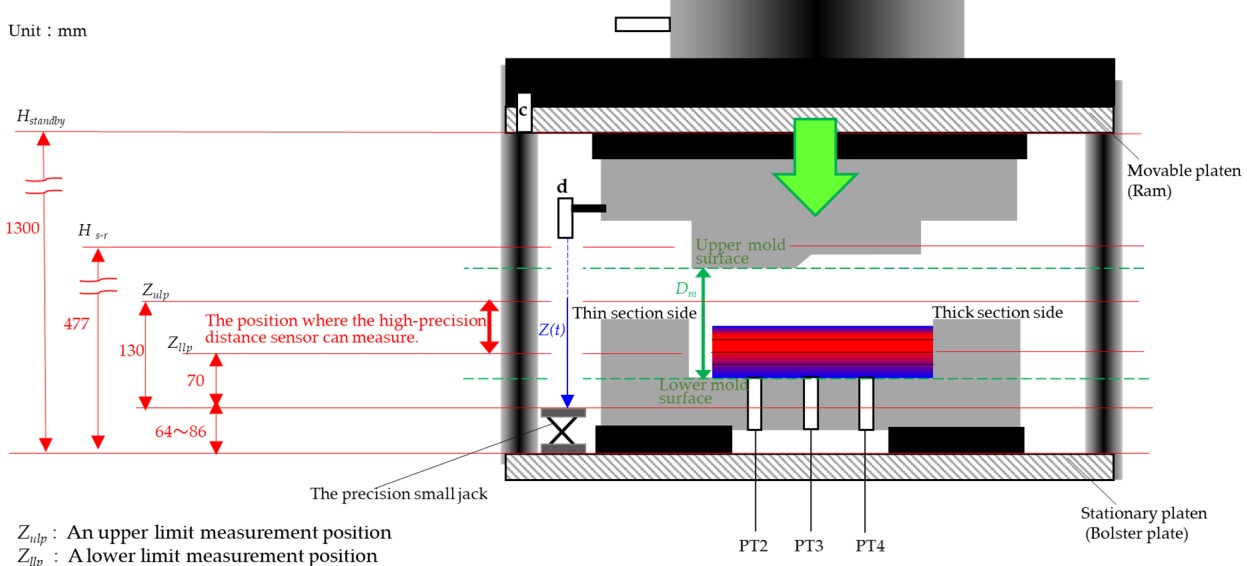

**Figure 6.** Schematic illustration of the upper mold position and measurement of distances between the upper mold surface and the lower mold surface during upper mold downward movement using the high-precision distance sensor (**c.** the long-distance sensor, **d.** the high-precision distance sensor).

Finally, after $t_{holding}$ had elapsed, the upper mold automatically moved up and a non-uniform thickness plate was demolded. Table 2 shows test cases for the compression molding conducted in the above experimental procedure. In test case 1 only, both the final stacked sheet and the mold cavity have flat shapes. In test case 2, only the mold cavity has a stepped shape. In test cases 3 to 5, both the final stacked sheet and the mold cavity have stepped shapes.

**Table 2.** Test cases for compression molding.

| Test Name | Stacking Pattern | Mold Type |
| --- | --- | --- |
| Test case 1 | Pattern A | Mold A |
| Test case 2 | Pattern A | Mold B |
| Test case 3 | Pattern B | Mold B |
| Test case 4 | Pattern C | Mold B |
| Test case 5 | Pattern D | Mold B |

## 5. Results and Discussion

Figure 7 shows a prior measurement result for the time-dependent changes in each temperature of surface and center in the thickness direction of a temporary 2-ply sheet during pre-heating and that in the temperature difference. The pre-heating time on the horizontal axis was set so that the time when the sheet was placed on the wire mesh panel, which is in a furnace controlled at 280 °C, was 0 s. The temperature near the door in the furnace decreased significantly from just before 0 s because the door was opened to place the 2-ply sheet into the furnace. The temperature decreased to approximately 60 °C. However, the door was closed quickly after placement so the temperature began to rise rapidly. In the short time period, although the temperatures of the surface and the center in the 2-ply sheet increased significantly, the increasing rate of the surface temperature was slightly faster than that of the center temperature. After that, as time advanced, the temperature of the 2-ply sheet approached that of the furnace, and the increasing rates gradually decreased. At the pre-heating time of 280 s, the temperature difference between the surface and the center increased to approximately 18 °C. However, after 280 s, the temperature difference between the surface and the center gradually decreased,

because the increasing rate of surface temperature became slower than that of the center temperature. At 476 s, not only the surface but also the center reached the target pre-heating sheet temperature $T_{pre\text{-}heat}$ of 270 °C. At this time, the temperature difference decreased to approximately 3.5 °C. Therefore, the author decided to pre-heat the 2-ply sheets for 476 s as preheating time $t_{pre\text{-}heat}$. Incidentally, the temperature difference became more significant when the number of stacked plies was increased to more than 2.

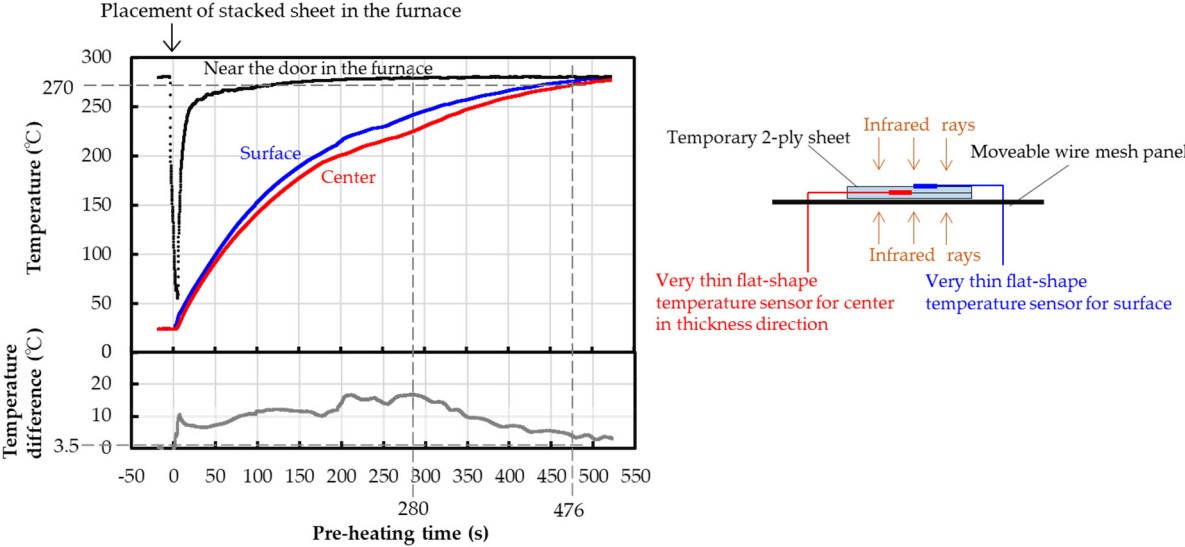

**Figure 7.** Prior measurement for temperature difference in the thickness direction of a temporary 2-ply sheet during pre-heating. The time when the sheet was placed on the wire mesh panel in the furnace was set to 0 s.

Figure 8 shows various data obtained through the monitoring system shown in Figure 2 when the final stacking pattern A and mold A were used as test case 1. The molding process time on the horizontal axis was set so that the time $t3$ at which the final 5-ply sheet was placed on the lower mold was 0 s. In other words, the time of the upstream process from the placement is expressed using the minus sign. First, with regard to the furnace temperature near the door, the temperature decreased significantly at approximately −494 s, because the door was opened to place the three sets of temporary 2-ply sheets into the furnace, controlled at 280 °C. The door was closed quickly after placement so that the temperature began to rise rapidly at approximately −487 s as a pre-heating start time $t1$. After the 2-ply sheets were pre-heated for the $t_{pre\text{-}heat}$ of 476 s, decided in the prior measurement, the door was re-opened in order to remove the 2-ply sheets from the furnace at a pre-heating stop time $t2$ of approximately −11 s. Therefore, the temperature decreased significantly again. It is possible to confirm whether pre-heating was performed according to the target conditions even after molding using the time-dependent changes in the furnace temperature and the following equation:

$$t_{pre\text{-}heat} = t2 - t1. \tag{2}$$

Subsequently, as shown in pattern A of Figure 5, only one sheet was eliminated from the three sets of the temporary 2-ply sheets, and a final 5-ply stacked sheet was quickly prepared. The 5-ply sheet was transferred to mold A, which was controlled at a target temperature of 160 °C.

Then, with regard to the lower mold surface temperature, the three pressure–temperature in-mold sensors showed values of 156 to 157 °C before the 5-ply sheet was placed on the lower mold. Immediately after the 5-ply sheet was placed on the lower mold, temperatures measured through these sensors began to increase because the temperature of the 5-ply sheet pre-heated is higher than that of the lower mold. That is to say, the moment when the 5-ply sheet was placed on the lower mold could be determined using the beginning

time of these temperature increases. Therefore, the molding process time *t*3 of 0 s was set to coincide with this beginning time. At approximately 11 s, the measured temperatures of all three sensors rapidly increased to approximately 167 °C. The author has called the increase from 0 s the first temperature increase and the increase from 11 s the second temperature increase.

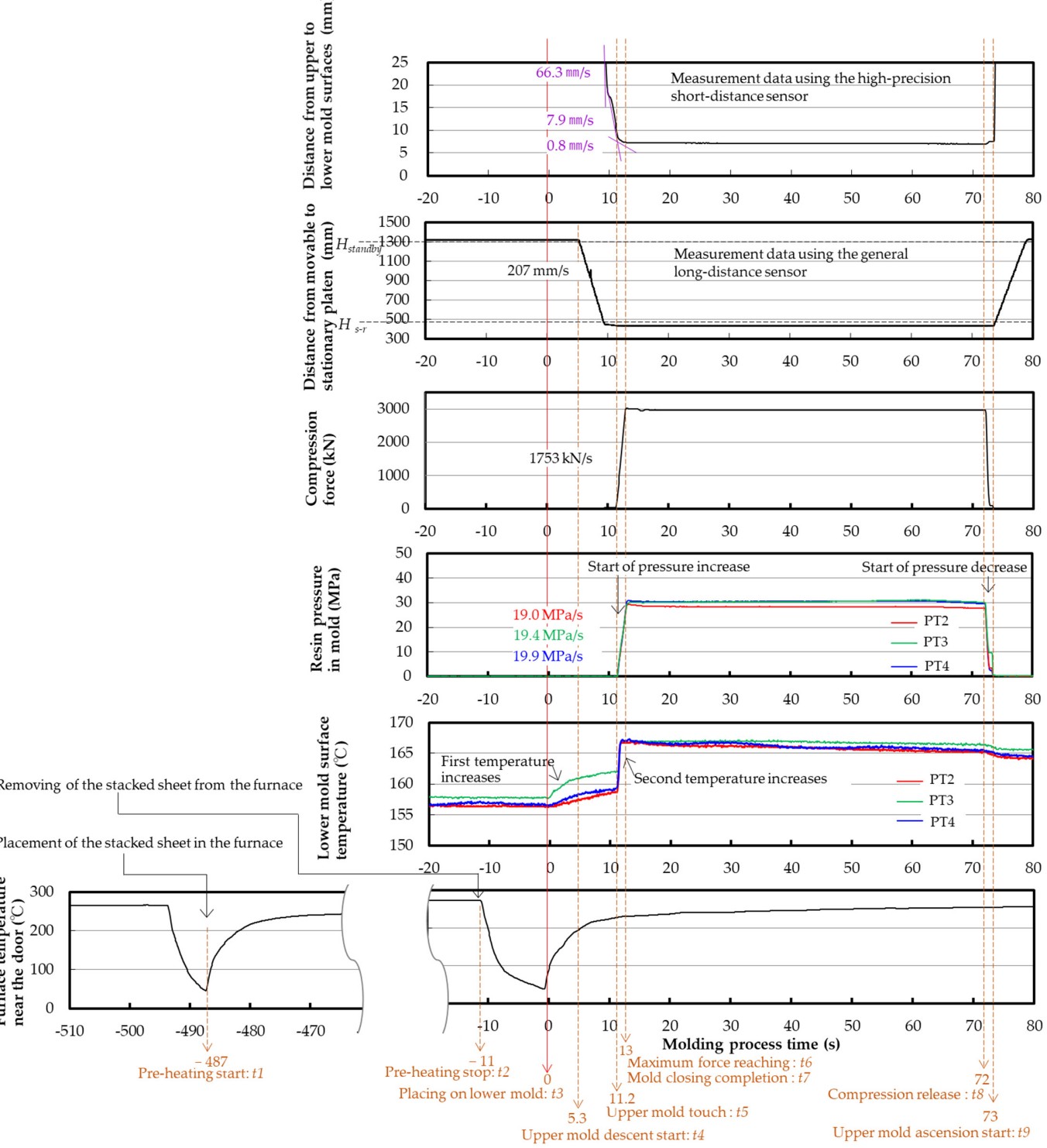

**Figure 8.** Logged data in the compression molding process using the test equipment and the monitoring system when the final stacking pattern A and the mold A were used as test case 1. The time when the sheet was placed on the lower mold was set to 0 s. ([Pressure increase rate] red: PT2, green: PT3, blue: PT4).

With regard to the resin pressure in the mold and the compression force, the three pressure–temperature in-mold sensors showed a value of 0 MPa until just before 11 s. At approximately 11.2 s, the measured pressures of all three sensors rapidly began to increase at a rate of 19.0 to 19.9 MPa/s. At approximately 13 s, each pressure reached approximately 30 MPa and then maintained its value. That is, the pressures measured at the three positions showed exactly the same time dependence. Here, the time at which the pressure began to increase was consistent with the beginning time of the second temperature increase. Therefore, the contact force between the sheet and the sensors increased so that the measured temperatures could rise. In other words, this time of 11.2 s can be determined as the time $t5$ when the upper mold started touching the sheet. Further, the compression force began to increase from 11.2 s at an increase rate of 1753 kN/s and reached the target of 3000 kN ($F_{max}$) at a maximum force reaching time $t6$ of 13 s. The beginning time of the compression force increase was the same as that of the pressure increase, and the time when the compression force became constant was the same as that when the pressure became constant.

Finally, with regard to distance from movable to stationary platen and distance from upper to lower mold surfaces, the platen distance showed a value of approximately 1300 mm ($H_{standby}$) until just before 5.3 s, when measured using the general long-distance sensor. From approximately 5.3 s as the start time $t4$ for the upper mold descending, the platen distance began to decrease at a speed of 207 mm/s. This was almost consistent with the set initial platen speed ($V_{initial}$). The upper mold motion just before completion of mold-closing could be observed in detail from the distance between upper and lower mold surfaces, which was calculated using Equation (1) and data measured through the high-precision short-distance sensor. The upper mold descended at a mold-closing speed of 66.3 mm/s until approximately 9.9 s and at a mold-closing speed of 7.9 mm/s from approximately 9.9 s to approximately 11.2 s. The upper mold finally moved down approximately 1 mm at a mold closing speed of 0.8 mm/s from approximately 11.2 s to approximately 13 s. The mold closing completion time $t7$ was approximately 13 s and was almost the same as the maximum force reaching time $t6$. That is, it was found that the speed immediately before the mold closing completion was changing in multiple stages within a short time. In measurements using only the general long-distance sensor, it was difficult to precisely observe the moving distance immediately before the mold closing completion.

After the holding pressure was applied for 60 s, the compression force began to decrease at a compression release time $t8$ of approximately 72 s, and the upper mold began to ascend for demolding at an upper mold ascension start time $t9$ of approximately 73 s. From these data, it is clearly possible to confirm whether or not the holding pressure was applied for the set $t_{holding}$, even after molding.

In addition, the time $t_{fs-t}$ required for both the final stacking and the transfer and the compression waiting time $t_{waiting}$ were able to be precisely calculated using the monitoring data and the following equations:

$$t_{fs-t} = t3 - t2, \tag{3}$$

$$t_{\text{waiting}} = t5 - t3. \tag{4}$$

The $t_{fs-t}$ was 11 s and $t_{waiting}$ was 11.2 s. Since $t_{fs-t}$ and $t_{waiting}$ have a major influence on the temperature change of the stacked sheet after pre-heating, these are important parameters that need to be monitored when moldability is examined. This monitoring method was able to distinguish between $t_{fs-t}$ and $t_{waiting}$ and save the data. The important point is that $t_{waiting}$, which was difficult to measure using conventional methods, can be evaluated with high precision. As a matter of course, it is possible to confirm whether each $t_{waiting}$ was appropriate for all molded products even after molding. Therefore, this monitoring method will also be useful for statistical process control (SPC) and product traceability in industry.

Figure 9 shows time changes of resin pressures in the molds that were obtained through the pressure–temperature in-mold sensors of the monitoring system shown in Figure 2 when each test case was conducted. In test case 1, the pressure time changes are shown only in the range of 0 to 25 s, including the pressure increase observed in Figure 8. In test cases 2 to 5, the pressure time changes are shown in the same time range as in test case 1. As described above, in test case 1 using the final stacked sheet with a flat shape and the mold cavity with a flat shape (i.e., the final stacking pattern A and the mold A), the pressures measured at the three positions indicated exactly the same behavior. In fact, at approximately 11.2 s, the measured pressures began to increase sharply at almost the same increase rates (19.0 to 19.9 MPa/s). From the data obtained at the sampling interval of 0.1 s, no clear difference was observed in the increase beginning time of each pressure. After that, each pressure reached approximately 30 MPa and then continued to indicate a constant value after approximately 13 s. By contrast, in test cases 2 to 5 using mold B, pressure time changes that were different from those in test case 1 were also observed.

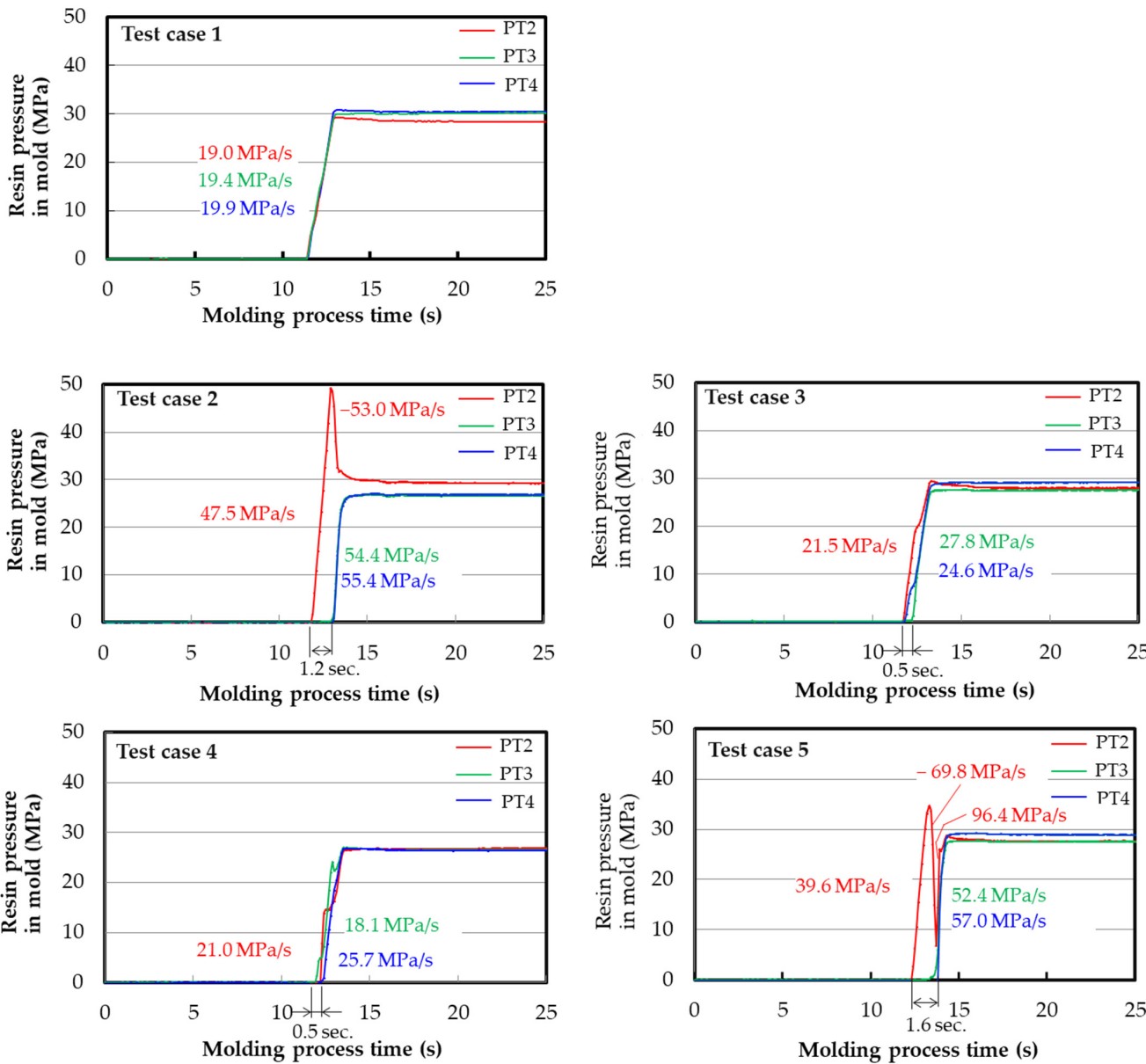

**Figure 9.** Time changes of resin pressures in the molds when each test case was conducted. ([Pressure increase and decrease rate] red: PT2, green: PT3, blue: PT4).

In test case 2, using a final stacked sheet with a flat shape and a mold cavity with a stepped shape (i.e., final stacking pattern A and mold B), at approximately 11.9 s, pressure in the thin section side (hereinafter referred to as PT2) began to increase sharply at an increase rate of 47.5 MPa/s and indicated a peak value of 48.5 MPa at approximately 13.0 s. Immediately afterwards, the PT2 decreased sharply at a decrease rate of −53.0 MPa/s and stabilized at approximately 30 MPa. In other words, PT2 showed significant pressure overshoot behavior. Two pressures in the stepped and thick section side (hereinafter referred to as PT3 and PT4, respectively) indicated almost the same behavior. At 13.0 and 13.1 s near the time when PT2 just peaked, PT3 and PT4 began to increase later than PT2, at somewhat higher rates of 54.4 and 55.4 MPa/s than PT2. After that, it stabilized at approximately 27 MPa, which is slightly lower than the stable value of PT2. PT3 and PT4 did not show pressure overshoot behavior. The difference (i.e., delay time) between the shortest and longest increase beginning times for the three pressure was 1.2 s. That is, different behavior also appeared at the three pressures. In comparison with test case 1, although only the increase beginning time of PT2 was almost the same, those of PT3 and PT4 were delayed and the increase rates of all three pressures were more than double.

In test case 3, in which a final stacked sheet with a stepped shape and a mold cavity with a stepped shape were used and the thick section of the sheet was placed on the thick section of the cavity (i.e., final stacking pattern B and mold B), within the range of 11.8 to 12.3 s, PT2, PT3, and PT4 began to increase at rates of 21.5 to 27.8 MPa/s and stabilized at approximately 29 MPa. Strictly speaking, the increase rates fluctuated slightly until each pressure reached a stable value. None of the three pressures showed overshoot behavior. The difference (i.e., delay time) between the shortest and longest increase beginning times for the three pressures was only 0.5 s. Essentially, almost the same behavior was demonstrated in the three pressures.

In test case 4, in which a final stacked sheet with a stepped shape and a mold cavity with a stepped shape were used and the thick section of the sheet was placed on the stepped section of the cavity (i.e., final stacking pattern C and mold B), within the range of 12.0 to 12.5 s, PT2, PT3, and PT4 began to increase at rates of 18.1 to 25.7 MPa/s and stabilized at approximately 27 MPa. Strictly speaking, the increase rates fluctuated until each pressure reached stable values. None of the three pressures showed overshoot behavior. The difference (i.e., delay time) between the shortest and longest increase beginning times for the three pressures was only 0.5 s. Essentially, almost the same behavior appeared in the three pressures.

In test case 5, in which a final stacked sheet with a stepped shape and a mold cavity with a stepped shape were used and the thick section of the sheet was placed on the thin section of the cavity (i.e., final stacking pattern D and mold B), at approximately 12.2 s, PT2 began to increase sharply at a rate of 39.6 MPa/s and indicated a peak value of 34.8 MPa at approximately 13.4 s. Immediately afterward, PT2 decreased sharply at a rate of −69.8 MPa/s and indicated a peak value of 6.8 MPa at approximately 13.7 s. Then, PT2 increased sharply again at a rate of 96.4 MPa/s and stabilized at approximately 28.8 MPa. In other words, PT2 showed anomalous pressure overshoot behavior in an upward and a downward direction. On the other hand, PT3 and PT4 indicated almost the same behavior. At approximately 13.4 s, when PT2 had just peaked in the upward direction, PT3 began to increase later at a higher rate of 52.4 MPa/s than PT2. After that, it stabilized at approximately 28 MPa. At approximately 13.8 s immediately after PT2 had just peaked in the downward direction, PT4 began to increase later at a higher rate of 57.0 MPa/s than PT2. After that, it stabilized at approximately 29 MPa. The difference (i.e., delay time) between the shortest and longest increase beginning times for the three pressure was as much as 1.6 s. That is, different behavior also appeared in the three pressures.

Here, the following was observed. From the results of test cases 1 and 2, it has been possible to confirm that resin pressure behavior can change significantly depending on the mold cavity shape even when the same final stacked sheet and the same molding condition are applied. From the results of test cases 2 to 5, it has been possible to confirm that resin

pressure behavior can change significantly depending on the final stacked sheet shape, even when the same mold cavity shape and the same molding condition are applied. That is to say, when at least one of the sheet shapes and the cavity shapes change, there are positions where the resin pressure changes locally and positions where the resin pressure hardly changes. In particular, PT2 of test case 2, in which a significant pressure overshoot occurred, indicated a peak value approximately 1.6 times higher than the stable value at other positions. PT2 of test case 5 showed two pressure increases and one pressure decrease in pressure within a short time range. The result is that two pressure overshoots were observed. The common points of test cases 2 and 5, in which the pressure overshoots were observed, are as follows:

- The increase rates of PT2, which showed pressure overshoot, were higher than those of all pressures in test cases 1, 3, and 4 which did not show pressure overshoot.
- PT3 and PT4, which did not show pressure overshoot, began to increase later than PT2, which showed pressure overshoot. In detail, at the time when PT3 and 4 began to increase, the pressure was almost the same as or immediately after the time when the peak value of PT2 was shown in the upward direction.
- The increase rates of PT3 and 4 were higher than those of PT2 in test cases 2 and 5 and those of PT3 and 4 in test cases 1, 3, and 4.
- Focusing on the final stacked sheet shape and the mold cavity shape, in order to fill completely in the mold cavity, the sheets in test cases 2 and 5 need to flow in a significantly more in-plane direction than those in test cases 1, 3, and 4.

The author considered resin flow mechanisms that might cause pressure overshoots. Figure 10 illustrates resin flow mechanisms that are considered to influence pressure overshoots, corresponding to pressure time changes. With regard to the pressure time changes in test case 2, at time i, the upper mold has not touched the final stacked sheet placed on the lower mold yet. At time ii, only on the thin section, the upper mold touches the sheet, and PT2 detects pressure. At time iii, as the upper mold moves further downward, and the resin flows in an in-plane direction. Since the stepped section in the mold cavity is also filled with resin, PT3 also begins to detect the pressure. The beginning of the in-plane flow seems to suddenly turn the increasing pressure detected in PT2 into a decrease. The author has considered that this is observed as a pressure overshoot. At time iv, since the thick section in the mold cavity is also filled with resin, PT4 also begins to detect the pressure. As mentioned above, the increase beginning times for pressures in PT3 and PT4 are 13.0 s and 13.1 s, respectively. That is, according to this mechanism, a transition from time iii to iv takes less than 0.1 s. Next, with regard to the pressure time changes in test case 5, at time i, the upper mold has not touched the final stacked sheet placed on the lower mold yet. At time ii, only on the thin section, the upper mold touches the sheet, and PT2 detects pressure. At time iii, as the upper mold moves further downward, the resin flows in an in-plane direction. Since the stepped section in the mold cavity is also filled with resin, PT3 also begins to detect the pressure. As in test case 2, the beginning of the in-plane flow seems to suddenly turn the increasing pressure detected in PT2 into a decrease. The author has considered that this is observed as a pressure overshoot in an upward direction. Furthermore, the resin flow amount can become slightly too much for the thick section, because the initial flow amount in-plane direction is considered to depend on the sheet amount of the fifth and sixth layers. At time iv, since the thick section in the mold cavity is also filled with resin, PT4 also begins to detect the pressure. At this moment, when there is an excessive amount of resin in the thick section, back-flow from the thick section to the thin section must occur. The beginning of the back-flow seems to suddenly turn the decreasing pressure detected in PT2 into an increase. The author has considered that this is observed as a pressure overshoot in a downward direction. The stacked sheets with very low thermal adhesion boundaries may indicate pressure overshoots significantly because they are more likely to move in an in-plane direction than bulk resins.

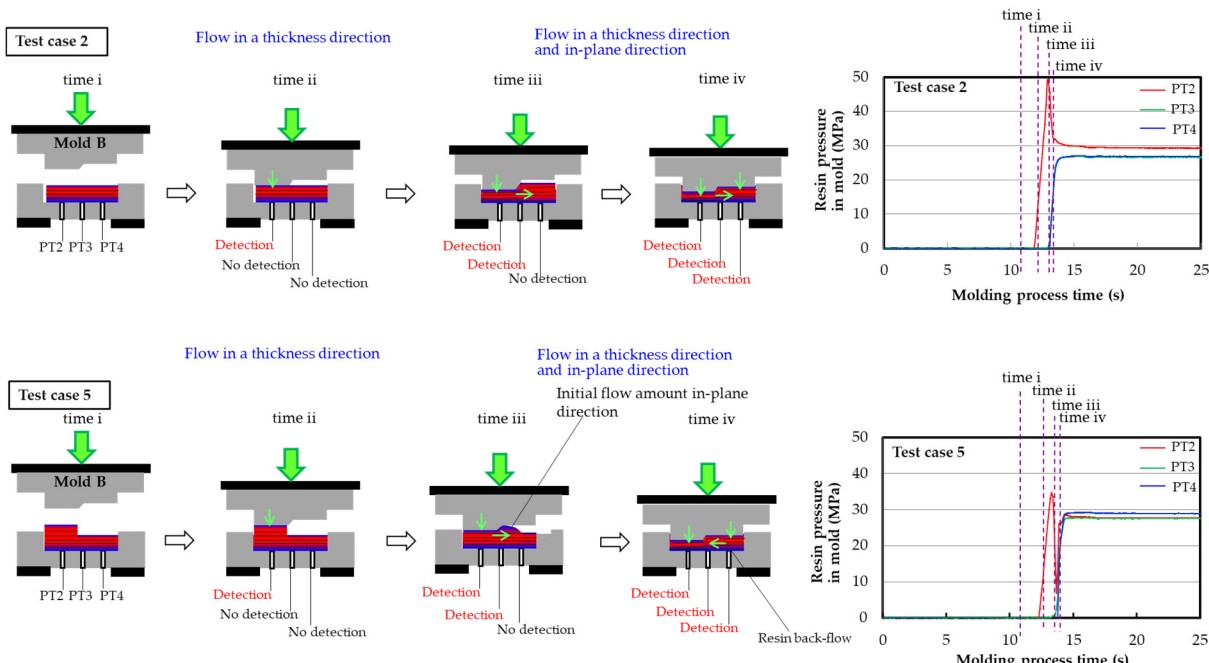

**Figure 10.** Resin flow mechanisms considered to influence the pressure time changes when each pressure overshoot occurs.

## 6. Conclusions

A monitoring system was built that can observe the compression molding process widely and precisely under non-isothermal conditions using glass-fiber-mat-reinforced thermoplastic (GMT) sheets. A feature of this system is to also use pressure–temperature sensors mounted on a compression mold that can simultaneously measure resin pressure and temperature on the lower mold at each local surface. In addition, both a general long-distance sensor and a high-precision short-distance sensor are used to monitor the movement of the moveable platen and the upper mold widely and precisely. Therefore, the time changes of the sheet behavior in the mold can be evaluated with high accuracy. Using this system, various data were logged from pre-heating to demolding after compression. The following are the study outcomes.

First, because of the pressure-temperature in-mold sensors, the moments when the final stacked sheets were placed on the lower mold were able to be detected through the change of the mold surface temperature, and the moments when the upper mold touched the final stacked sheet were also able to be continuously detected through the beginning of pressure increase. Therefore, the compression waiting time from sheet placement on the lower mold to the start of sheet compression by the upper mold could be precisely calculated. That is, unlike conventional monitoring methods, this method was able to distinguish the compression waiting time from the final stacking and transfer times.

Finally, using this monitoring system, the influences of final stacked patterns and mold cavity shapes on time changes in resin pressures were examined. The resin pressure behavior can change significantly depending on the final stacked sheet shape and the mold cavity shape, even when the same molding condition is applied. On the other hand, depending on the specific shape difference, the resin pressure behavior may hardly change. In cases where the resin pressure behavior changed significantly, the pressure in the thin section of the mold cavity for the non-uniform thickness plate began to increase earlier than those in the stepped and the thick sections, and pressure overshoots were clearly observed. In test case 5, pressure overshoots in both an upward and a downward direction were also observed. In cases where overshoot was observed, the increasing rates in pressure were higher than those in cases where it was not observed. Due to the geometrical commonality of the test cases where pressure overshoots were observed, in comparison with the final stacked sheet shape and the mold cavity shape, the sheets needed to flow significantly in

a more in-plane direction in order to fill the mold cavity completely. On the other hand, due to the geometrical commonality of the test cases where pressure overshoots were not observed, the sheets did not need to flow in a significantly in-plane direction in order to completely fill the mold cavity.

This monitoring system can be used not only for compression molding of GMT sheets but also for those that include a pre-heating process. That is, this could also be applied to the LFT-D molding process with a heating machine to inhibit a temperature decrease of heated LFT strands. The results for pressure overshoots will also be useful, not only for compression moldings of GMT sheets but also for those of LFT. The author has recommended that sheet stacking patterns be determined by considering mold cavity shapes so that excessive pressure is not applied to molds. In other words, a thick section of stacked sheets should be placed on the local thick area of the mold cavity. Conversely, a thin section of stacked sheet should be placed on the local thin area of the mold cavity.

**Funding:** This research received no external funding.

**Conflicts of Interest:** The authors declare no conflict of interest.

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
