# Peer review of "Study of Monitoring Method and Melt Flow Behavior in Compression Molding Process Using Thermoplastic Sheets Reinforced with Discontinuous Long-Fibers"

_jcs, doi:10.3390/jcs5020050_

Round 1

Reviewer 1 Report

Dear author!

Currently, many varieties of processes for the production of parts made of polymeric composites are used in various industries. Moreover, thermoplastics are increasingly being used even for critical high-load parts. Therefore, it would be useful for a wide range of readers to familiarize themselves with the implementation problems of the process under consideration in more detail. I recommend that the author provide a schematic illustration of the process under study, indicating the main components, equipment and the duration stages of the process in the Introduction, not in main part of the article.

The description of the hydraulic machine device is unnecessarily detailed and distracts from its main features.

When considering the conditions for conducting experiments with different patterns of stacking layers, the question arises: why did the author investigate cases 4, 5 (patterns C, D). Their necessity is not obvious. It is advisable to answer this question at the very beginning of the Experimental Methods section, or describing Figure 4.

When reading the experimental conditions, it is very important to understand the behavior of the viscosity of the thermoplastic used at the temperatures described. Unfortunately, the temperature dependence of the viscosity for Polyamide 6 is not given in the article.

The experimental conditions are described in detail in the article. However, one of the most important tasks of a scientific article is to generalize the results and formulate recommendations based on them. Such generalization and recommendations are absent in the main text and in the conclusions.

The reviewed article contains practically important information regarding the manufacturing of parts made of  thermoplastic composites of the type under consideration. However, the text of the article requires some corrections necessary to improve its readability, the possibility of using its results, and increasing their scientific significance.

Author Response

January 29th, 2021

Dear Reviewer 1,

I am submitting a revised version of my manuscript (Manuscript ID: jcs-1080381), which was originally submitted on December 31st, 2020. It has been revised in line with the reviewers' comments. Please also find attached responses to the reviewers' comments.

Since I was asked to correct the English, My revised version was checked by a native English translator. This is for readability and scientific significance, and any new facts are not added other than responses to the reviewer's requests.

I would like to thank the editor and reviewers for their helpful comments and suggestions.

Yours sincerely,

Masatoshi Kobayashi (Dr. Eng.)

Reviewer 2 Report

In this article, the authors have built their own compression molding process and sequences to fabricate GMT sheets. All the experiments were systematically tested and evaluated.

The so called "Compression-waiting-time" method was presented to calculate precise profile data by the authors.

The compression molding process is important in industrial fields and their applications, so this work provides a good and new measure for precise profile data calculation and monitoring of GMT sheets. In my opinion, this work can be published in the journal in this current form provided that poor English grammar is improved by English editor. 

Author Response

January 29th, 2021

Dear Reviewer 2,

I am submitting a revised version of my manuscript (Manuscript ID: jcs-1080381), which was originally submitted on December 31st, 2020. It has been revised in line with the reviewers' comments. Please also find attached responses to the reviewers' comments.

Since I was asked to correct the English, My revised version was checked by a native English translator. This is for readability and scientific significance, and any new facts are not added other than responses to the reviewer's requests

I would like to thank the editor and reviewers for their helpful comments and suggestions.

Yours sincerely,

Masatoshi Kobayashi (Dr. Eng.)
